ecology

hydrothermal vent, Southwest Indian ridge, population connectivity, scaly-foot snail, *Chrysomallon*

**Authors for correspondence:**
Chengjun Sun
e-mail: csun@fio.org.cn
Pei-Yuan Qian
e-mail: boqianpy@ust.hk

†Equal contribution.

# Nearest vent, dearest friend: biodiversity of Tiancheng vent field reveals cross-ridge similarities in the Indian Ocean

Jin Sun[1,2,†], Yadong Zhou[3,†], Chong Chen[4],
Yick Hang Kwan[2], Yanan Sun[2], Xuyang Wang[5],
Lei Yang[6], Ruiyan Zhang[3], Tong Wei[2], Yi Yang[2],
Lingyun Qu[1], Chengjun Sun[1] and Pei-Yuan Qian[2]

[1]Marine Bioresource and Environment Research Center, First Institute of Oceanography, Ministry of Natural Resources, Qingdao 266061, People's Republic of China
[2]Department of Ocean Science, Division of Life Science and Hong Kong Branch of the Southern Marine Science and Engineering Guangdong Laboratory, The Hong Kong University of Science and Technology, Hong Kong, People's Republic of China
[3]Key Laboratory of Marine Ecosystem Dynamics, Second Institute of Oceanography, Ministry of Natural Resources, Hangzhou, People's Republic of China
[4]X-STAR, Japan Agency for Marine-Earth Science and Technology (JAMSTEC), 2-15 Natsushima-cho, Yokosuka, Kanagawa 237-0061, Japan
[5]State Key Laboratory of Ocean Engineering, Shanghai Jiao Tong University, Shanghai 200240, People's Republic of China
[6]Marine Survey Research Center, First Institute of Oceanography, Ministry of Natural Resources, Qingdao 266061, People's Republic of China

JS, 0000-0001-8002-6881; CC, 0000-0002-5035-4021;
YHK, 0000-0003-4107-5650

Biodiversity of hydrothermal vents in the Indian Ocean, particularly those on the Southwest Indian Ridge (SWIR), are still relatively poorly understood. The Tiancheng field on the SWIR was initially reported with only a low-temperature diffuse flow venting area, but here we report two new active areas, including a chimney emitting high-temperature vent fluids. Biological sampling in these new sites doubled the known megafauna and macrofauna richness reported from Tiancheng. Significantly, we found several iconic species, such as the scaly-foot snail and the first *Alviniconcha* population on the SWIR. Tiancheng shares a high proportion of taxa with vents on the Central Indian Ridge (CIR) and lacks a number of key taxa that characterize other vents investigated so far on the SWIR. Population genetics of the scaly-foot snail confirmed

this, as the Tiancheng population was clustered with populations from the CIR, showing low connectivity with the Longqi field. Unlike the previously examined populations, scales of the Tiancheng scaly-foot snail were coated in zinc sulfide, although this results only from precipitation. The close connection between Tiancheng and CIR vents indicates that the dispersal barrier for vent endemic species is not the Rodriguez Triple Junction as previously suggested but the transformation faults between Tiancheng and Longqi, warranting further studies on deep currents in this area to resolve the key barrier, which has important implications for biological conservation.

## 1. Introduction

Since their initial discovery in 1977, deep-sea hydrothermal vents sustaining extremely lush biological communities for the deep ocean have attracted great attention from scientists and the general public alike [1]. The majority of studies have, however, been carried out in the Pacific or Atlantic oceans, leaving the Indian Ocean vent biological communities the least explored among the three major oceans. So far, only nine active hydrothermal vent fields have been discovered in the Indian Ocean (figure 1a). The first vent was the Kairei field on the Central Indian Ridge (CIR) discovered in 2000, very close to the Rodrigues Triple Junction where three mid-ocean ridges meet [2]. Afterwards, a number of other vents have been discovered on the CIR including the Edmond [3], Solitaire, Dodo and Onnuri fields [4,5]. Carlsberg Ridge is the northern extension of the CIR, where the Wocan field was recently discovered [6]. From 2007 onwards, the Southwest Indian Ridge (SWIR) has been the focus of a number of research cruises resulting in the discovery of three active vent fields in two areas, with Longqi and Duanqiao fields in the western part and Tiancheng in the eastern part [7,8]. The third ridge, Southeast Indian Ridge (SEIR), has long been neglected with no active vents being visually confirmed, but finally the first one, Pelagia field, was found in 2017 [9].

Multivariate analyses using megafauna and macrofauna community structure data have revealed a total of 11 biological provinces worldwide for hydrothermal vents, with vents in the Indian Ocean clustering into a single province [10,11]. In addition, considering that larvae of vent-endemic fauna often disperse along flank jets or 'highway' currents on the ocean ridge axis [12], Indian Ocean vents have been hypothesized to be a dispersal corridor between the western Pacific and the Atlantic ocean, through SEIR and SWIR [13]. The evidence for this hypothesis is so far rather mixed [14], and more studies on the vent-endemic fauna in SEIR and SWIR are required to better test the species distribution and also the vent biogeography [15]. This is largely hindered by the fact that no vents have been explored on the SEIR and that the Longqi field in the central portion of the SWIR remains the only well-characterized SWIR vent field in terms of venting activity or megafaunal and macrofaunal biodiversity [7,8,16].

Many peculiar animal species have only been found in Indian Ocean vent fields, and understanding how their populations are connected can help shed light on their dispersal and on how vent taxa we see today evolved through dispersal and isolation. The most peculiar one is the scaly-foot snail, *Chrysomallon squamiferum*, which is thus far known from four vent fields on the CIR and the SWIR, including the Kairei, Solitaire, Longqi and Duanqiao fields [17]. Like many vent endemic animals, it houses sulfur-oxidizing endosymbiotic bacteria and relies on them for nutrition [18,19]. Uniquely among snails, it carries hundreds of dermal sclerites on its foot which are infused with biomineralized iron sulfide in the population discovered in Kairei [20]. Recently, it was revealed that the scaly-foot snail mediates the biomineralization process by actively pumping out sulfur component through numerous nano-channels in the scale, and environmental iron ions react with the sulfur to form species of iron sulfide, including pyrite and greigite [21]. Different populations have varying levels of metal mineralization linked to the composition of vent fluid, ranging from none in Solitaire to crystallized iron sulfide in Kairei [21]. Its habitat is also threatened due to potential future mining activity, and it has become the first deep-sea species to be Red Listed by the IUCN (as an Endangered species) due to risks of deep-sea mining [22], after the collecting for the present study has taken place.

So far, a population genetic analysis has revealed low connectivity between scaly-foot snail from the Longqi field and the two CIR populations [23], which remains the only population genetics study in the Indian Ocean vents to study population connectivity across two ridges. However, considering the Longqi vent field is the only population sampled on the SWIR and is 2300 km away from Kairei on the CIR, it would be of great interest to include other populations between the two. As the Rodriguez Triple Junction where the CIR, SWIR and SEIR meet has been proposed as the key dispersal barrier due to supposedly rare cross-ridge dispersal events [23], the hypothesis would be that populations will be better connected within each ridge compared to between ridges. However, no major rift offsets are

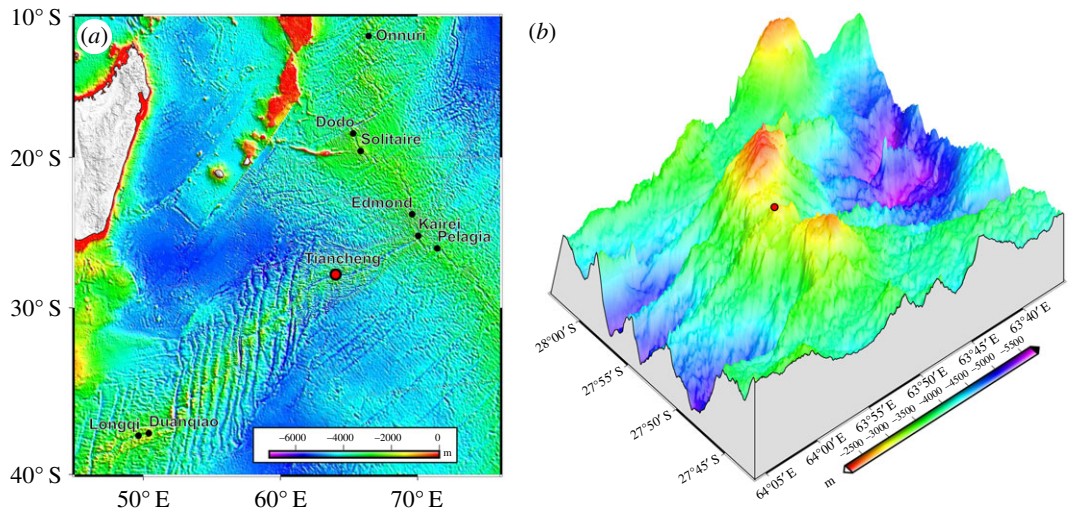

**Figure 1.** Location of the Tiancheng hydrothermal vent field shown (*a*) in the Indian Ocean with other known hydrothermal vent sites labelled as reference, and (*b*) on the Tiancheng Seamount showing using 3D bathymetry.

present between the Kairei/Edmond (CIR) and Tiancheng (SWIR), even at the Rodriguez Triple Junction [24]. An alternative hypothesis involves major transformation faults that are present between Tiancheng and Longqi on the SWIR, such as Galieni and Atlantis II, that offset the ridge axis by more than 100 km between the two vent fields [24]. These may act as significant dispersal barriers and limit the connectivity between Tiancheng and Longqi, while the quasi-continuum between Tiancheng and Kairei may result in the two sites being connected.

The Tiancheng vent field is in a perfect geological location for testing these hypotheses, as it is located on the eastern part of SWIR, between Longqi (approx. 1700 km to the west) and Kairei (approx. 650 km to the east). It was originally reported by Tao *et al.* [25], and the biological composition was initially described by Zhou *et al.* [8]. However, both studies have found only a single low-temperature diffuse flow area and relatively few megafaunal and macrofaunal taxa. In April 2019, during leg III of DY52nd cruise by R/V *Dayangyihao*, we located two further active areas within the vent field, which included another low-temperature diffuse flow site and also a highly active chimney site—the first in the eastern part of the SWIR. The new chimney site was also found to house a new population of the scaly-foot snail, among a variety of other species previously unreported at this vent field. This study aims to update and provide a detailed account of the megafaunal and macrofaunal community in the Tiancheng vent field, as well as to use population genetics to examine its role in vent fauna connectivity in the Indian Ocean.

## 2. Material and methods

Biological samples were collected from three venting sites in the Tiancheng vent field (figure 1*b*), located on the slope of the Tiancheng Seamount, i.e. the JL-87 diffuse flow area (this area was previously reported by Zhou *et al.* [8]), the TC-2 diffuse flow area, and the Tiantang (meaning 'heaven' in Chinese) chimney (electronic supplementary material, figure S1) with suction sampler by the remotely operated vehicle (ROV) *Sea Dragon III* during the COMRA DY52$^{nd}$ cruise of R/V *Dayangyihao* in the Southwest Indian Ocean in April 2019. Upon arrival on-board the ship, animals were sorted and immediately fixed in pure ethanol or −80°C deep freezer.

Animals were identified to the lowest possible level morphologically, and confirmed with genetic barcoding where possible. For each species, DNA was extracted using the DNeasy Blood & Tissue Kits (Qiagen). The mitochondrial cytochrome *c* oxidase I (COI) gene was amplified and sequenced using the primers from Leray *et al.* [26]. All amplified PCR products were sequenced and then used for phylogenetic analysis with reference sequences from [23,27,28]. All DNA sequences were aligned in MEGA 7 for all orthologue genes [29].

Three dominant megafauna species (the scaly-foot snail, the deep-sea mussel *Bathymodiolus marisindicus* and the crab *Austinograea rodriguezensis*) were selected for population genetic analyses. Haplotype network analysis was computed by the median-joining Network in PopART v. 1.7, using the obtained COI sequences [30]. The analysis of pairwise-$F_{st}$ values was calculated by using Arlequin v. 3.5 [31]. In addition, the genetic

diversity, which included the number of haplotypes, haplotypic diversity and nucleotide diversity of the individuals in Tiancheng were computed in DnaSP v. 6 [32]. For comparative purposes, additional specimens of *Bathymodiolus marisindicus* were sequenced from Longqi (15 individuals) and Duanqiao (10 individuals) vent fields, using samples collected by the human-occupied vehicle (HOV) *Jiaolong* during the COMRA DY35[th] cruise (see Zhou *et al*. [8] for details on sample collection).

Non-metric multidimensional scaling (nMDS) in PRIMER v. 6 (PRIMER-E, Plymouth UK) was used to assess the inter-field similarities of the biological community structure of vents from the CIR, SWIR, East Scotia Ridge (ESR) and the Mid-Atlantic Ridge (MAR) following the methods of Zhou *et al*. [8]. This includes updated data for Tiancheng from the present research added to those from previous studies [8,10,16,27]. The dataset is based on species-level presence/absence, with indeterminate species of the same genus being assigned to different taxonomic units if collected from different vent fields.

Since the minerals that deposit on the sclerites of *Chrysomallon squamiferum* reflect the biochemical reaction of the snail itself with the chemical property of the vent fluid [21], to examine and assess the mineralization and elemental composition of the sclerites from individuals collected from Tiantang chimney in the Tiancheng hydrothermal vent field, scanning electron microscopy (SEM) with energy dispersive X-ray spectrometry (EDS) analyses was undertaken using a Hitachi TM3000 EDS equipped with a Bruker QUANTAX EDS (JAMSTEC). The surface and the cross-section through a scale was analysed using the elemental mapping function of the software QUANTAX 70, focusing on key elements including iron, sulfur, zinc, oxygen, nitrogen and carbon. This automatically overlaid the EDS elemental maps on an SEM image of the analysed areas.

# 3. Results

Faunal assemblages at three venting sites, as well as in the periphery, have been characterized: JL-87 diffuse flow site (figure 2), TC-2 diffuse flow site (figure 3) and the Tiantang chimney (figure 4). The two diffuse flow venting sites, JL-87 and TC-2, are more than 150 m apart, suggesting a relatively large area under hydrothermal influence comparable with the Kairei and Edmond fields [3].

## 3.1. Habitat characterization

The JL-87 diffuse flow site (63.9233° E, 27.8508° S, 2682 m in depth) is located on a ridge close to the summit of the Tiancheng Seamount. Faunal aggregates were found to mainly surround a cave-like structure with a triangular opening formed by large basaltic rubbles. Transparent shimmering fluids were observed rising from the floor inside the cave. However, we were not able to determine their physical and chemical characteristics as the small opening of the cave did not allow the ROV to enter for fluid collection. Shimmering water emission found near the cave opening was measured to be 14°C in temperature, compared to the background water of approximately 2°C [8].

The TC-2 diffuse flow site (63.9219° E, 27.8505° S, 2729 m in depth) is characterized by an oval depression with a diameter of approximately 5 m. The hydrothermal influence here was initially indicated through the presence of high faunal abundance and the presence of some vent endemic fauna. Then numerous diffuse flow regions were detected on the margin of the depression. No prominent accumulation of hydrothermal fluid precipitation (i.e. chimney structure) was found, with only exposed rocks and faunal assemblages being observed.

The Tiantang chimney (63.9227° E, 27.8508° S, 2705 m in depth) is also located on the slope, about 60 m west of the JL-87 diffuse flow area. It is a composite of two active chimney structures, surrounded by basalt. Topped with branching structures, the chimneys were less than 2 m in height and arranged in parallel to the slope. Dense dark grey fluids and intensified discharging were observed at several smoker dischargers at the top, indicating high-temperature hydrothermal emissions. The hydrothermal precipitates consisted of accumulations of at least two minerals, sphalerite and anhydrite. A narrow diffusing area, featuring transparent shimmering water and dense aggregation of vent fauna, centres on the base of the chimneys.

## 3.2. Faunal composition and spatial distribution

Altogether, 23 morphospecies were found at three sites in the Tiancheng field (table 1). Molluscs, with six species, and crustaceans, with five representatives, were the two groups with the highest level of diversity. A total of 12 species (10 invertebrates, table 1) are here reported for the first time from the

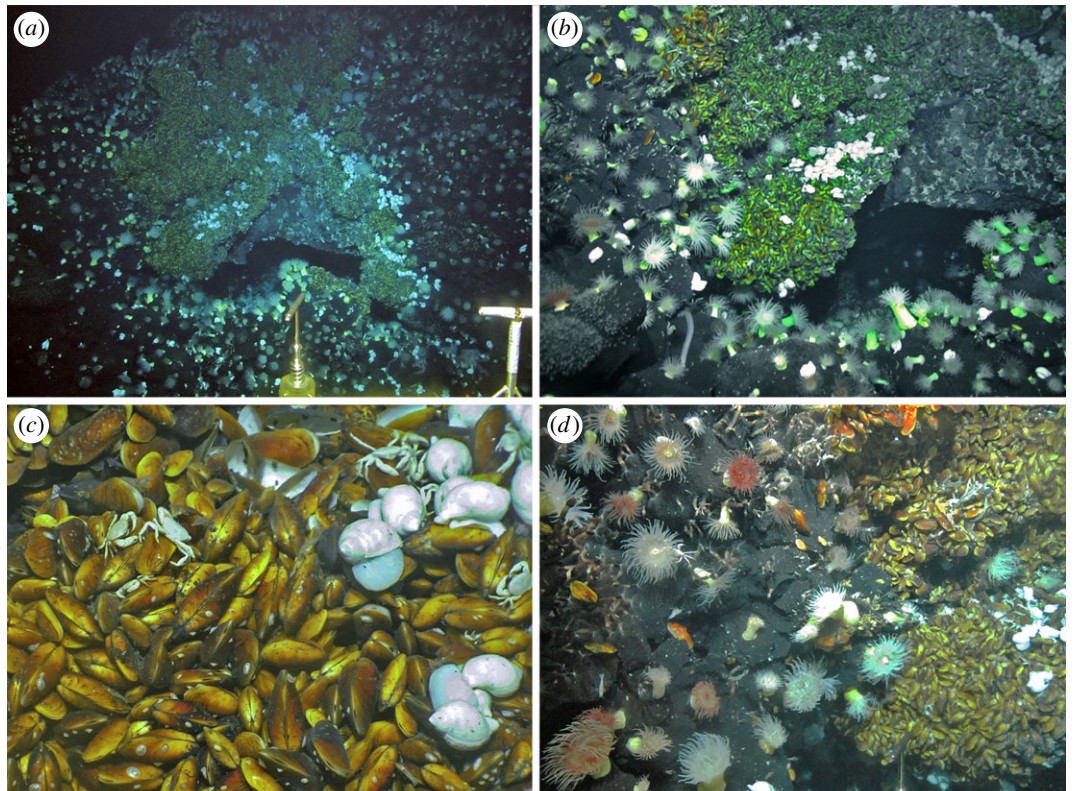

**Figure 2.** JL-87 diffuse flow site. (*a*) Overview showing the cave-like basalt feature. (*b*) Close-up of the cave opening showing the centre of venting activity and Type-I assemblages. (*c*) Close-up of Type-II assemblages dominated by the deep-sea mussel *Bathymodiolus marisindicus*. (*d*) Close-up of the transition area from Type-II to Type-IV assemblages, with a narrow strip of Type-III assemblage dominated by the stalked barnacle *Neolepas marisindica* in between.

Tiancheng field, including the scaly-foot snail *Chrysomallon squamiferum*, the polychaete worm *Ophryotrocha* sp., a dirivultid copepod and the sea cucumber *Chiridota* sp., etc. The scaly-foot snail from Tiancheng has a distinct appearance from other known populations on the CIR and SWIR [17], with a reddish-brown overlay on dirty-white scales and a light brown shell.

At the three venting sites, an array of faunal assemblages formed by a varied combination of organisms was distributed in zonation along the gradient, apparently determined by the distance from the vent fluid sources: (1) Type-I Assemblage, with dense mixtures of *Austinograea* crabs and the scaly-foot snail forming dense aggregations covering the low-temperature fluid orifices, with the shrimp *Mirocaris indica* also commonly seen; (2) Type-II Assemblage, with thick layers of the deep-sea mussel *Bathymodiolus marisindicus* and limpets *Pseudorimula* sp. and *Eulepetopsis* sp. living on their shells in abundance, occupies areas peripheral to Type-I assemblages; (3) Type-III Assemblage dominated by dense clumps of stalked barnacles (*Neolepas marisindica*) together with small separate patches of the conoidean gastropod *Phymorhynchus* sp. and individuals of the sea cucumber *Chiridota* sp. and large *Marianactis* sea anemones, fills the remaining gaps in the area influenced by visible fluids; (4) Type-IV Assemblage is located in the peripheral area dominated by two morphotypes (white and purple) of *Marianactis* sea anemone; (5) Type-V Assemblage is also in the periphery but is instead dominated by dense fields of unidentified small sea anenomes. Type-IV and Type-V peripheral assemblages (figure 5) cover wide areas of basalt rock surrounding the three venting sites, and *Phymorhynchus* sp., fishes (family Ophidiidae and cf. *Polyacanthonotus* sp.), *Austinograea* crabs and sea stars (family Pterasteridae) were occasionally spotted.

In JL-87 (figure 1), the fluid originates inside the cave, and the Type-I assemblage can only be seen from the cave opening, but is too distant for a clear image to be captured or specimens to be sampled. The community outside the cave shows an abrupt transition from Type-II to Type-IV assemblages, with Type-III being only present in narrow strips between them. Due to the discontinuous fluid emission orifices, the community at the TC-2 diffuse flow site is divided into several small patches, with each composed of Type-I, II or III in successive zonation. Chimney structures issuing high-temperature fluids are hallmarks of the Tiantang chimney site, but no fauna was spotted aggregating on chimney walls. Type-I assemblages fill the long narrow space surrounding the base of the chimneys. Type-II and III assemblages successively inhabit

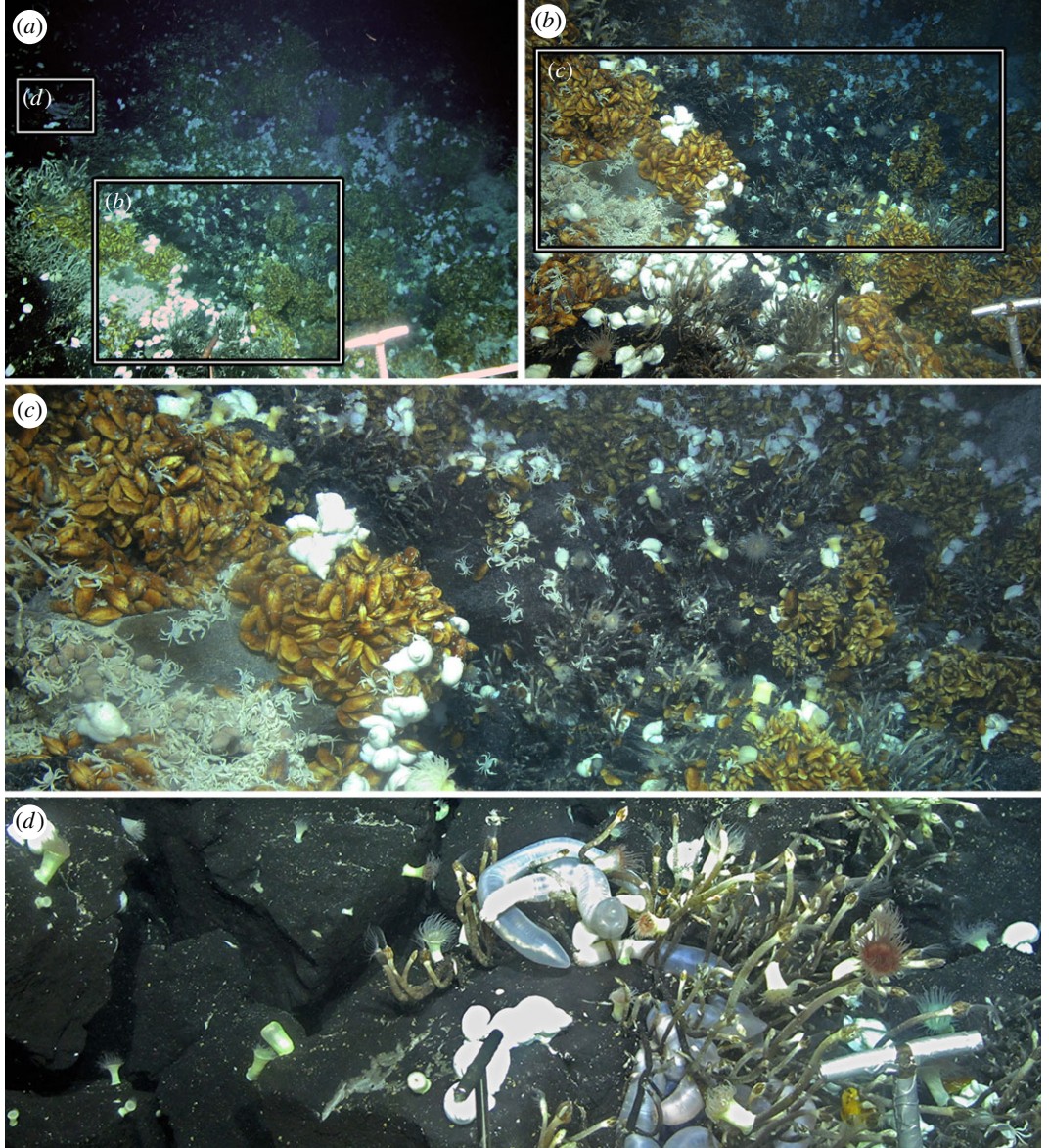

**Figure 3.** TC-2 diffuse flow site. (*a*) Overview of the site. (*b*) Close-up showing Type-I, II and III assemblages. (*c*) Zonation transiting from Type-I to Type-II to Type-III assemblages. (*d*) Type-IV assemblages on the periphery showing sea anemones and the sea cucumber *Chiridota* sp.

narrow bands further away from the chimneys. The peripheral areas immediately adjacent to active areas of the three sites were all represented by a ring of Type-IV assemblages, marking the transition to reduced influence from hydrothermal activity. Further out, a wide expanse of Type-V assemblages connects all three active sites, indicating that they are probably best treated as parts of the same vent field.

Community structure analysis with nMDS has revealed that the Tiancheng biological community is in between the Longqi (SWIR) and Kairei / Solitaire fields (CIR) (figure 6, raw species presence/absence data can be accessed in electronic supplementary material, table). The whole biological community in the Indian Ocean vent fields forms a major cluster, supporting the theory that they are one biological province.

## 3.3. Population genetics

Population connectivity of three dominant species (i.e. the scaly-foot snail *Chrysomallon squamiferum*, the deep-sea mussel *Bathymodiolus marisindicus* and the crab *Austinograea rodriguezensis*; figure 7) in the Tiancheng vent field was investigated by sequencing the partial sequence of the mitochondrial COI gene and comparing with the published data [23,27,28]. For *A. rodriguezensis*, the sampled

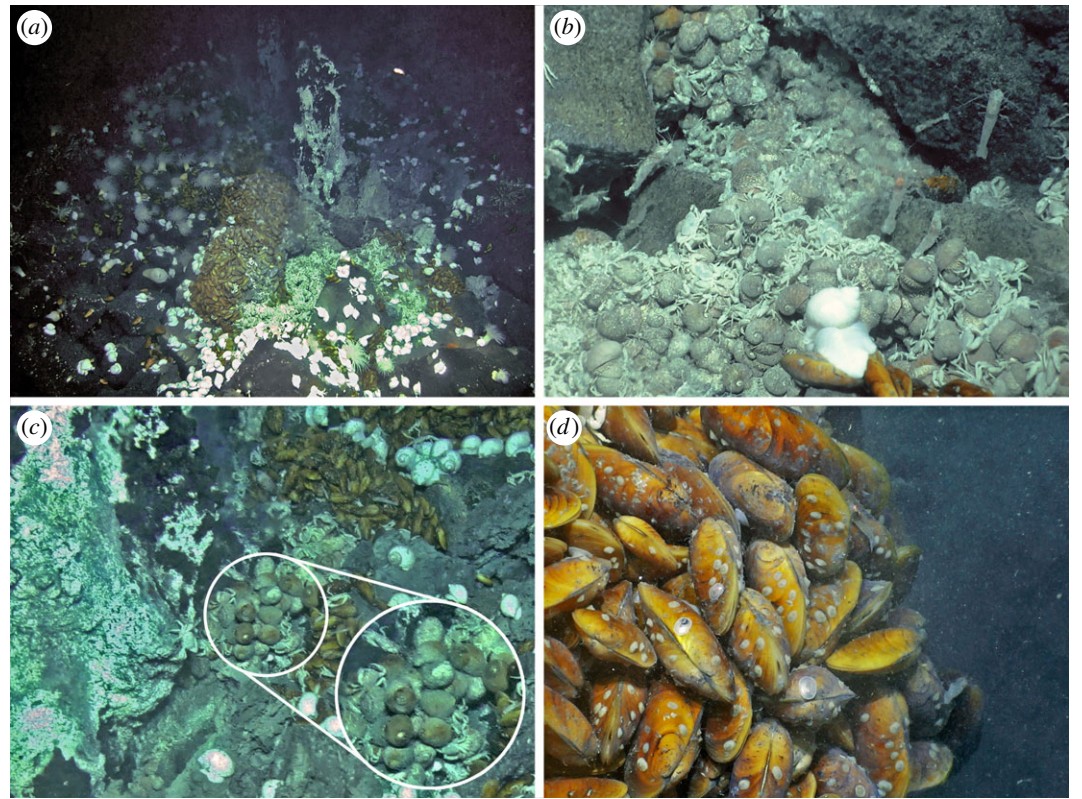

**Figure 4.** Tiantang chimney site. (*a*) Overview of the site with two volatile, active chimneys with dense megafaunal assemblages at the base. (*b*) Type-I assemblage dominated by the scaly-foot snail and the crab *Austinograea rodriguezensis*. (*c*) A cluster of *Alviniconcha* cf. *marisindica* snails near Type-I assemblages. (*d*) Type-II assemblages dominated by the mussel *Bathymodiolus marisindicus*.

populations were from Tiancheng, Kairei and Solitaire. For *B. marisindicus*, populations included were from Kairei, Edmond, Duanqiao, Longqi and Tiancheng; with samples from the three SWIR vents being newly sequenced in this study. For *C. squamiferum*, populations included were from Kairei, Solitaire, Tiancheng and Longqi (electronic supplementary material, tables S1 and S2).

In general, for *A. rodriguezensis* and *B. marisindicus*, pairwise $F_{st}$ value and Wright's exact tests among the population indicated that there are no significant genetic differences among all populations sampled (electronic supplementary material, table S1). On the other hand, significant population differentiation was found for *C. squamiferum* between Longqi and all other populations with $F_{st}$ value ranging from 0.30345 to 0.40506 (table 2). Although a significant *p*-value was recovered between Tiancheng and Kairei, the $F_{st}$ value is only 0.07838, much lower than that between Longqi and the rest.

Haplotype networks (figure 8) were also constructed. The haplotype network of *A. rodriguezensis* was relatively simple with a star-burst shaped network with one slightly dominant haplotype shared equally among the three populations sampled. This confirms that there is a lack of population differentiation among the three vents. The haplotype network of *B. marisindicus* was highly diversified, but there is no population forming one distinct cluster, also indicating high population connectivity among all of the known populations in the Indian Ocean. By contrast, the haplotype of *C. squamiferum* from Longqi formed a different cluster from the rest of the haplotypes and with only a single shared haplotype between Longqi & Kairei [23]. The hapolotypes obtained from the Tiancheng population were mixed and indistinguishable from haplotypes from the CIR populations.

## 3.4. Analyses of scales of the Tiancheng scaly-foot snail

The scales of the Tiancheng scaly-foot snail were revealed under the microscope to have a dirty white base coloration overlaid by a sparsely distributed, reddish-brown layer. SEM-EDS analyses of the scales' surface layer (figure 9; electronic supplementary material, figure S2) indicated a high level of sulfur, zinc and oxygen in the reddish-brown deposits, while iron was clearly not present in quantity. The scale's actual organic matrix below the surface layer had a distinctly different elemental composition rich only in oxygen, carbon and nitrogen, but depleted in iron or zinc. Sulfur was

**Table 1.** Taxa list and definitions of faunal assemblages in Tiancheng vent community. '+', '++' and '+++' in black colour indicate the presence of the species in low, medium and high abundance, respectively; '+++' in red colour means dominant species of the assemblages. Bold species names indicate newly recorded taxa from Tiancheng.

| phylum | family | species | assemblage I | II | III | IV | V | unassigned taxa | COI accession no. |
|---|---|---|---|---|---|---|---|---|---|
| Cnidaria | Actinostolidae | *Marianactis* sp. Tiancheng | | + | ++ | +++ | + | | MH202753 |
| | | **Small unidentified sea anemone** | | + | ++ | ++ | +++ | | — |
| Annelida | Polynoidae | *Branchipolynoe longqiensis* | | ++ | | | | | MT067560 |
| | | **Branchinotogluma bipapillata** | | + | | | | | MT067561 |
| | | Polynoidae sp. 3 *sensu* Zhou *et al.* [8] | | + | | | | | — |
| | Dorvilleidae | **cf. Ophryotrocha sp.** | | | | | | + | — |
| Mollusca | Mytilidae | *Bathymodiolus marisindicus* | | +++ | + | + | + | | MT052804–MT052822 |
| | Peltospiridae | **Chrysomallon squamiferum** | +++ | | | | | | MN840594–MN840613 |
| | Raphitomidae | *Phymorhynchus* sp. Tiancheng | + | ++ | ++ | + | + | | MT067563 |
| | Provannidae | **Alviniconcha cf. marisindica** | ++ | ++ | | | | | — |
| | | **Desbruyeresia cf. marisindica** | | | | | | + | MT067562 |
| | Neolepetopsidae | *Eulepetopsis* sp. | | +++ | | | | | MT067566 |
| | slit limpet | *Pseudorimula* sp. | | +++ | | | | | MT067564 |
| Arthropoda | Eolepadidae | *Neolepas marisindica* | | ++ | +++ | + | | | MH202767 |
| | Bythograeidae | *Austinograea rodriguezensis* | +++ | ++ | + | + | + | | MN821028–MN821042 |
| | Alvinocarididae | *Mirocaris indica* | ++ | + | + | | | | — |
| | Galathidae | **Galathidae indet.** | | | | | | + | — |
| | Dirivultidae | **Dirivultidae indet.** | | | | | | + | — |
| Holothuroidea | Chiridotidae | **Chiridota sp.** | | + | + | + | | | — |
| | Pterasteridae | **Pterasteridae indet.** | | | | | | + | — |
| Chordata | Ophidiidae | Ophidiidae indet. | | | | | | + | — |
| | Notacanthidae | **cf. Polyacanthonotus sp.** | | | | | | + | — |
| | Macrouridae | **Macrouridae indet.** | | | | | | + | — |

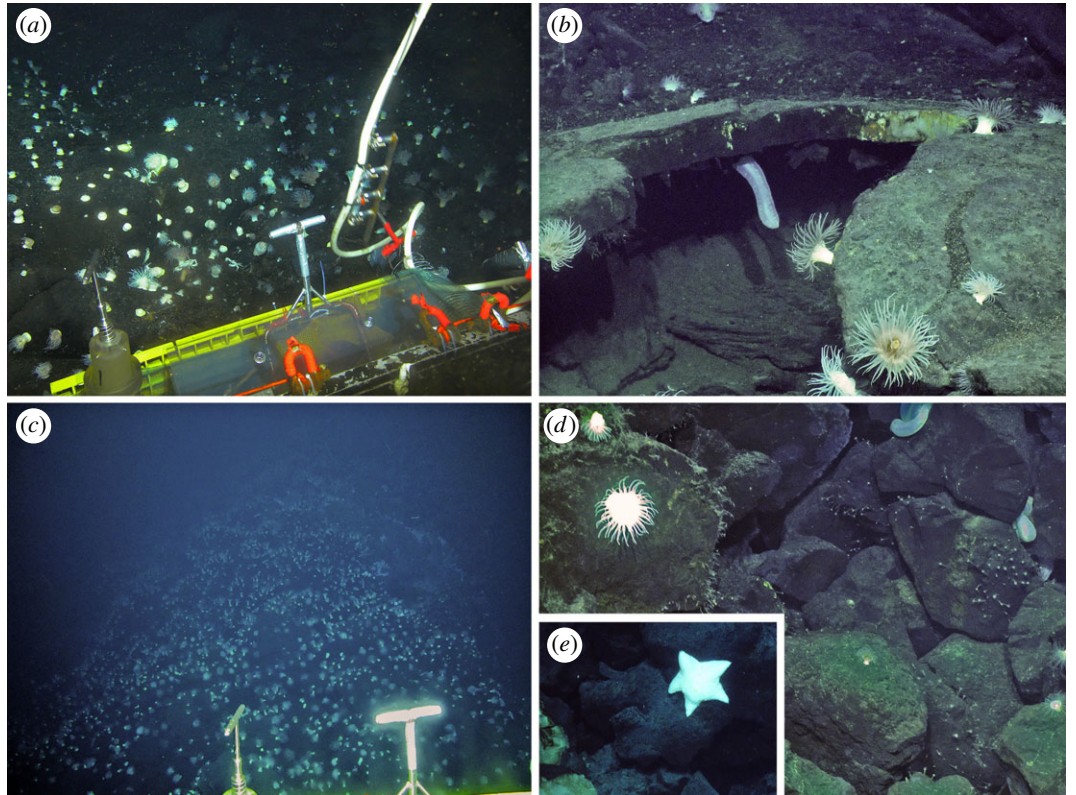

**Figure 5.** Periphery area between the three main active sites. (*a*) Type-IV assemblages dominated by *Marianactis* sp. large sea anemones. (*b*) Close-up of Type-IV assemblages showing *Marianacris* sp. and a sea cucumber *Chiridota* sp. overhanging from the roof of a basalt cave-like structure. (*c*) The transition from Type-IV to Type-V assemblages dominated by unidentified small sea anemones. (*d*) Type-V assemblage with small sea anemone and *Chiridota* sp. between basalt. (*e*) A pterasterid seastar.

present but in lower levels than the surface deposit layer. No signature of infused iron sulfide or other minerals were present below the surface deposits, indicating a lack of biomineralized mineral particles inside the scales' organic matrix. As the distribution of oxygen on the surface layer did not co-localize with zinc and sulfur, the outer reddish-brown layer is deduced to be most likely zinc sulfide. Also, in the areas of the scale lacking the reddish-brown layer, zinc was not detected at high levels.

# 4. Discussion

## 4.1. Biodiversity of the Tiancheng field

Results from the present study provide a comprehensive overview of the Tiancheng hydrothermal vent field, improving our understanding of the extent of its hydrothermal activities as well as its fauna from the first report by Zhou *et al*. [8]. Previously, only one diffuse flow area (JL-87) had been characterized, with a total of 11 species recorded, based on limited observation and sampling efforts during just one manned submersible dive. The current study adds two active areas including the discovery of the first high-temperature smoker chimney on the eastern half of the SWIR, confirming the predictions by Chen *et al*. [33]. Our new data also double the known number of megafauna and macrofauna species from the vent field, report the discovery of a new population of the scaly-foot snail *Chrysomallon squamiferum*, and provide the zonation pattern which has been described for a number of other vents [34].

## 4.2. Scaly-foot snail from Tiancheng

The Tiancheng scaly-foot snail, unlike other previously reported populations, has a reddish-brown coloration due to zinc sulfide deposits on the surface of its sclerites. From the scaly-foot snail's first discovery in the Kairei field which hosts a population with black scales infused with iron sulfide nanoparticles [20], biomineralization has been an important aspect of research carried out on this

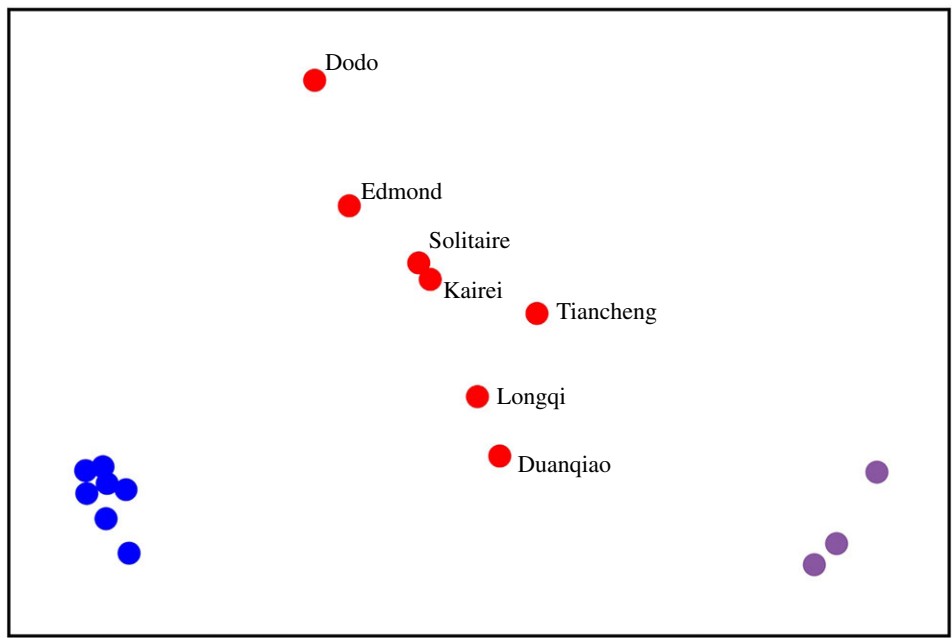

**Figure 6.** Two-dimensional non-metric multidimensional scaling (nMDS) plot of inter-field Soresen Index for 17 hydrothermal vents on CIR (red), SWIR (red), ESR (purple) and MAR (blue) at the species level. MDS ordination stress: 0.01.

species [35]. Now we know that the Kairei population makes iron sulfide using a unique mechanism, i.e. the sulfur deposited into the scales by the snail itself reacts with iron ions diffused in from the very iron-enriched vent fluid. The discovery of a population with scales lacking iron but still enriched in sulfur in the Solitaire field [5], as well as the Longqi population with amorphous iron sulfide coating [20,21,36] indicated that iron is not necessary for the animal's survival, but the sulfur deposition activity may be an adaptation to remove sulfur metabolites (likely from its thioautotrophic endosymbionts) from its body [21]. Our SEM-EDS results showing a lack of zinc sulfide infused within the scale's organic matrix indicate that the zinc sulfide on the Tiancheng population is not the result of biomineralization but instead likely natural deposition from the vent fluid. This is corroborated by the irregular distribution of the reddish-brown zinc sulfide on the scale's surface, and the fact that removing this layer results in a whitish surface similar to scales of the Solitaire population [5]. The presence of some sulfur in the organic matrix hints that the Tiancheng population may also be depositing excess sulfur into the scales. Although zinc sulfide has not been previously reported as a major mineral associated with the scaly-foot snail, it is commonly formed in hydrothermal vents with zinc-enriched endmember fluids [37]. This hints that the metal contents of the endmember fluid of the Tiancheng field may be rather different from those of Kairei, Solitaire or Longqi.

## 4.3 Implications on vent connectivity in the Indian Ocean

A key observation from this improved understanding of the biological community at Tiancheng is its striking similarity to CIR vents, unlike other previously known active vent fields (i.e. Longqi and Duanqiao) on the SWIR. Geographically, Tiancheng is much closer to the Rodriguez Triple Junction and Kairei field than to Longqi and Duanqiao, located approximately 1700 km to the east of Longqi. The assumption that larvae of many vent endemic animals use the deep flank jet currents on the mid-ocean ridges may suggest that cross-ridge dispersal events are less likely to occur [12,15]. Tiancheng, however, is located in an interesting geological setting where there are no major transform faults between it and Kairei on the CIR, and instead several major transformation faults are present on the SWIR between it and Longqi [24]. Our results, therefore, indicate that Rodriguez Triple Junction is much less significant of a dispersal barrier compared to major transform faults along the SWIR, which may be expected at least for animals considered to disperse through deep currents (e.g. gastropods with lecithotrophic larvae). Tiancheng actually lacks many presumably lecithotrophic taxa known solely from Longqi and Duanqiao such as the peltospirid snails *Gigantopelta aegis*, *Dracogyra subfusca* and *Lirapex politus* [38,39]. Conversely, a number of taxa previously only known from CIR vents were found at Tiancheng. These include the crab *Austinogrea rodriguezensis* and

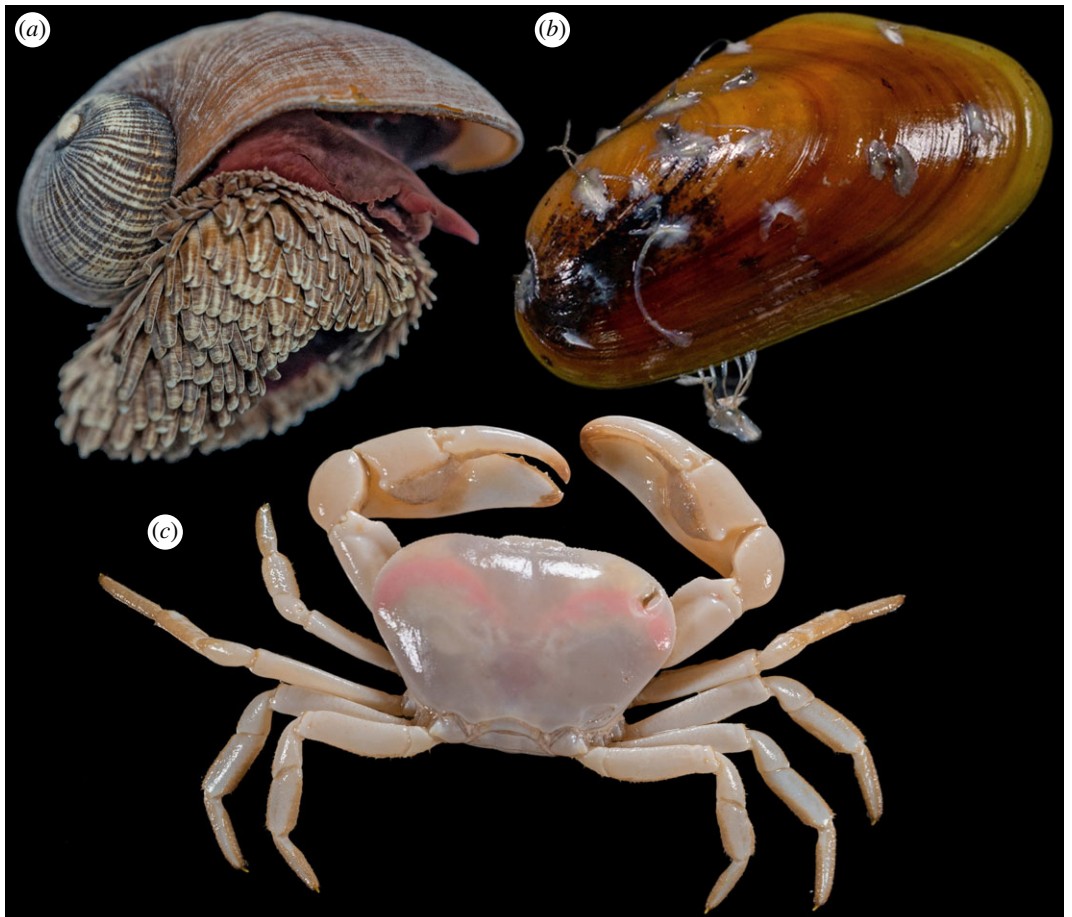

**Figure 7.** Representative individuals of the three dominant megafauna species selected for population genetic analyses. (*a*) The scaly-foot snail *Chrysomallon squamiferum*, shell length approx. 40 mm. (*b*) The mussel *Bathymodiolus marisindicus*, shell length approx. 45 mm. (*c*) The crab *Austinograea rodriguezensis*, carapace width approx. 40 mm. Photos courtesy of Dr Chenggang Liu.

**Table 2.** A summary table of pairwise $F_{st}$ value for *Chrysomallon squamiferum*.

|  | Kairei | Solitaire | Tiancheng | Longqi |
|---|---|---|---|---|
| Kairei |  |  |  |  |
| Solitaire | −0.00932 |  |  |  |
| Tiancheng | 0.07838*** | −0.00707 |  |  |
| Longqi | 0.30345*** | 0.30591*** | 0.40506*** |  |

***$p$-value < 0.01.

the provannid snail *Alviniconcha*. Although geographically the *Alviniconcha* in Tiancheng is most likely *A. marisindica*, we were unable to collect specimens and must refer to it as *Alviniconcha* cf. *marisindica* for the time being. The *Desbruyeresia* species found in Tiancheng is the same as previously reported from Longqi [8] (99% pairwise similarity in COI sequence), although no genetic information is available from the type locality of *Desbruyeresia marisindica*, Kairei, and thus we refer to it as *D.* cf. *marisindica* for now. Furthermore, many taxa shared between Tiancheng and the other SWIR vents were widely ranging species also known from CIR vents, such as the mussel *Bathymodiolus marisindica*. These findings collectively indicated that the community structure of Tiancheng is more similar to CIR vents than to Longqi or Duanqiao further west on the SWIR. The nMDS analysis placed Tiancheng among the other Indian Ocean vents, showing that in a broader scale these vents are all part of a single global vent biogeographic province.

Tiancheng's link with CIR vents is further strengthened by our population genetics results. The scaly-foot snail was previously reported to exhibit low connectivity between the Longqi, SWIR population and two

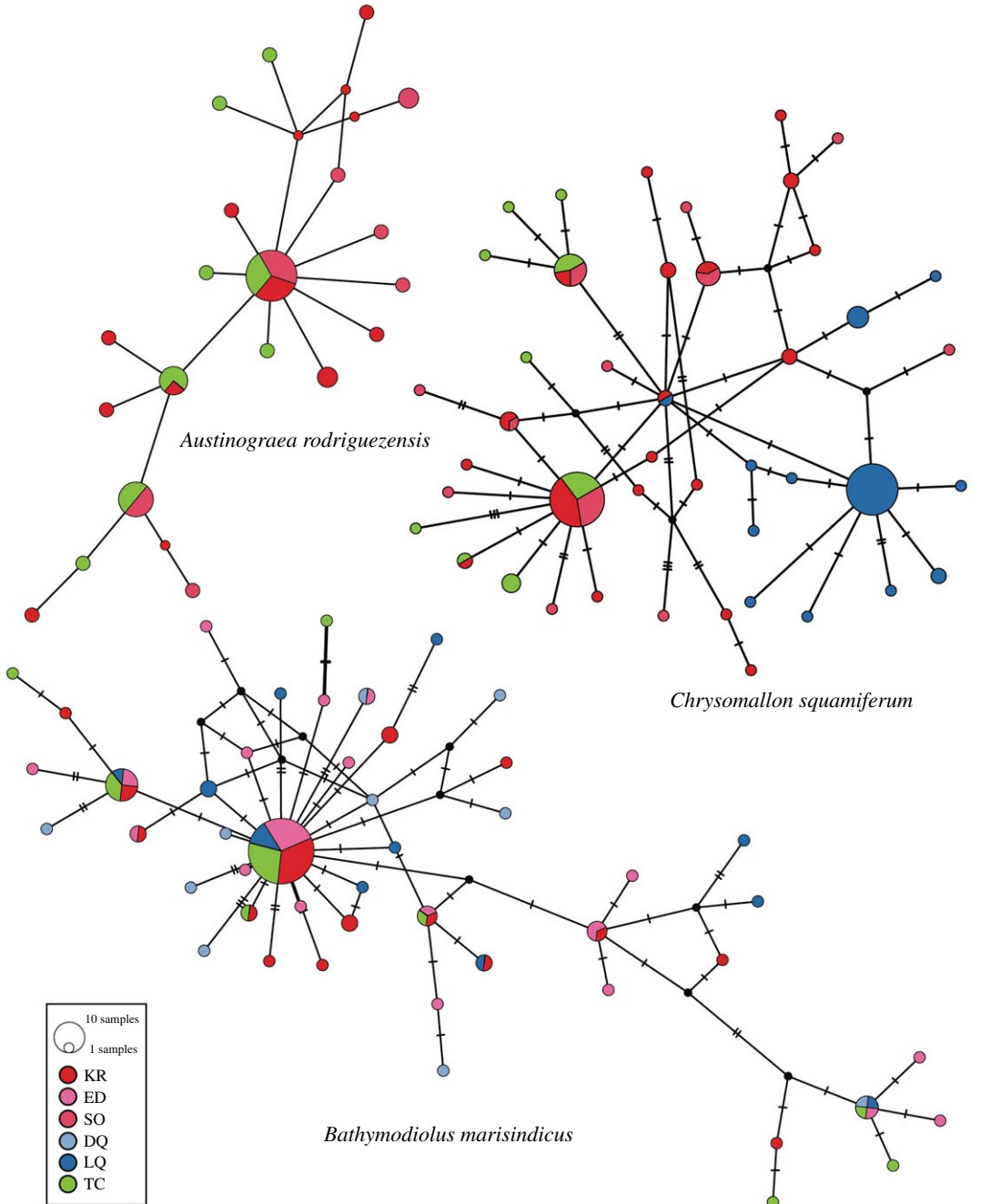

**Figure 8.** Haplotype networks of three dominant species including *Chrysomallon squamiferum*, *Bathymodiolus marisindica* and *Austinograea rodriguezensis*.

populations on the CIR that were panmictic [23]. Without any populations between the 2300 km distance between Longqi and Kairei, the authors could only predict that the genetic barrier was most likely to be the Rodriguez Triple Junction. By adding the Tiancheng population, we demonstrated that the Tiancheng population is well connected with CIR populations and also exhibited low connectivity with Longqi. This is also supported by the fact that the crab *Austinograea rodriguezensis* also showed panmixia across Tiancheng and two populations on the CIR. These reinforce the idea that the key dispersal barrier for these animals is not the Rodriguez Triple Junction but the transformation faults that lies in between Tiancheng and Longqi on the SWIR [12], warranting future studies.

Population genetics of *Bathymodiolus marisindicus*, on the other hand, indicated a panmictic condition across all six sampled vent fields ranging from Longqi, SWIR to Solitaire, CIR without any population structure. This is perhaps unsurprising because *Bathymodiolus* have long been known to have extremely high dispersal capabilities, with their planktotrophic larvae that in one case have been collected in surface water [40,41]. *Bathymodiolus marisindicus* is a member of the *Bathymodiolus septemdierum* complex

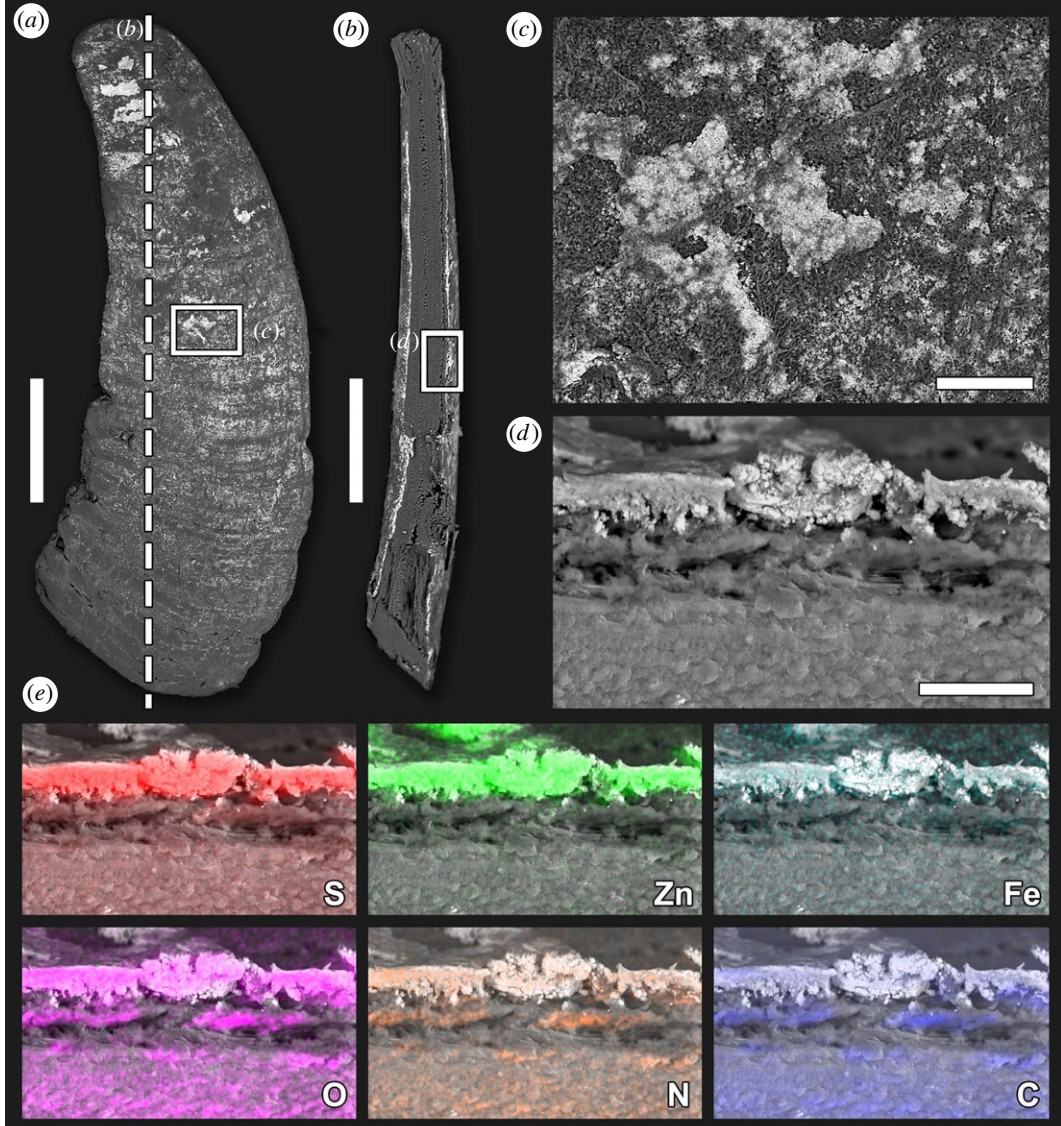

**Figure 9.** Scanning electron microscopy (SEM) and elemental dispersive X-ray spectrometry (EDX) of the scale of the Tiancheng scaly-foot snail. (*a*) Overview of a scale. (*b*) A cross-section through the scale cut as indicated by the dotted line. (*c*) Close-up of the scale surface showing unevenly distributed mineral deposits. (*d*) Close-up of the cross-section showing mineral deposit layer on the scale. (*e*) EDX mapping of the same area showing the distribution of sulfur (S), zinc (Zn), iron (Fe), oxygen (O), nitrogen (N) and carbon (C), brightness indicates relative concentration. Scale bars, (*a,b*) 1 mm; (*c*) 100 µm and (*d*) 20 µm.

(i.e. *Bathymodiolus septemdierum*, *B. brevior*, *B. elongatus* and *B. marisindicus*) which together ranges from Izu-Ogasawara Arc in Japan to Longqi, SWIR. These 'species' show little genetic divergence across this enormous range with significant population differentiation only being detected when multiple nuclear and mitochondrial genes were combined, leading some authors to suggest they may be better treated as a single species [28].

As such, it is likely that the dispersal barrier between Tiancheng and Longqi is confined to deep waters and does not affect *B. marisindicus* which presumably is able to disperse through shallow surface waters. The fact that the egg of the scaly-foot snail is negatively buoyant also supports this hypothesis [27]. This would also explain why the peltospirid snails *Gigantopelta aegis*, *Dracogyra subfuscus* and *Lirapex politus* were not found in Tiancheng, as their larvae were inferred to be lecithotrophic based on the protoconch [38], suggesting limited dispersal ability through deep currents. The absence of *Alviniconcha* from Longqi and Duanqiao may require additional explanations, however, as its larvae are supposedly planktotrophic [42]. Nevertheless, it is not known at what depths *Alviniconcha* larvae disperse. As a number of fauna specific to Longqi are close congers with Antarctic vent species on the ESR (e.g. *Kiwa* yeti crabs, *Gigantopelta* snails) [10,38], the Antarctic Circumpolar Current transporting larvae from ESR to the southwestern

portion of SWIR possibly have significantly influenced the community structure at Longqi & Duanqiao [43]. Collection of specimens and genetic information are required to identify and examine the connectivity of *Alviniconcha* population in Tiancheng with those on the CIR, which would be an important focus for future visits to the vent. The absence of the vent shrimp *Rimicaris kairei* in Tiancheng, like Duanqiao [8], is also an interesting characteristic since this species is extremely abundant in CIR vent fields and is also known from Longqi but present only in very low abundance [8,16]. Since Tiancheng is in-between Kairei and Longqi, this indicates that the presence or absence of *R. kairei* is not limited by larval dispersal capabilities but instead the local environment, likely linked to larval settlement preferences. The different mineralization present on the scales of the Tiancheng scaly-foot snail compared to other sites may reflect such differences in the local environment, especially in the vent fluid composition. In 'extreme' environments such as hydrothermal vents, taxa are often locally adapted to a narrow range of environments and the environment may in turn select for which taxa may be present in environmental filtering, which may be an important factor determining species distributions in the Indian Ocean hot vents. Understanding the interaction between biology and the environment in this sense is of great importance in biological conservation of these sites, for example, protection of sites with a wide range of environment variation may be required in order to conserve the full species diversity.

# 5. Conclusion

In the present study, we presented a comprehensive account of the chemosynthesis-based megafaunal and macrofaunal community structure in the Tiancheng vent field, the easternmost active vent field currently confirmed on the SWIR. We reported the first volatile venting chimneys in this area, and about a dozen species previously unrecorded for this vent field including a new population of the scaly-foot snail. This new population is characterized by reddish-brown zinc sulfide deposits, which do not appear to result from biomineralization but are likely owing to natural accumulation from vent fluid. Comparing the community structure with other vents indicated that Tiancheng was more similar to CIR vents rather than with the two other known vents (Longqi and Duanqiao) on the SWIR further west. Population genetic results detected substantial population structure between Longqi and Tiancheng but not between Tiancheng and CIR, supporting the hypothesis that the significant fracture zones between Longqi and Tiancheng on the SWIR present key dispersal barriers for most vent animals but not the Rodriguez Triple Junction, enabling cross-ridge dispersal events to occur between CIR and Tiancheng. These results warrant future studies into the specific impacts of each fracture zone and differing impacts to different taxa with varying dispersal strategies. Currently, hydrothermal vents are being targeted for deep-sea mining of massive sulfide deposits and the majority of known active vent fields in the Indian Ocean, including Tiancheng, are within mining exploratory license areas issued by the International Seabed Authority. To ensure the sustainable management of these 'treasures of the deep', we must strive to document and understand the biodiversity at these vents and the mechanisms shaping this diversity, before it becomes too late.

Ethics. The faunal collections were conducted in International Waters. All applicable international, national and/or institutional guidelines for the care and use of animals were followed by the authors. Research animals collected were non-cephalopod invertebrate animals and no experiments with animals were conducted in this study.

Data accessibility. The sequences were deposited at NCBI, with the accession number of MN840594-MN840613 for *Chrysomallon squamiferum*, MN821028-MN821042 for *Austinograea rodriguezensis*, MT052802-MT052845 for *Bathymodiolus marisindicus* and MT067560-MT067566 other species.

Authors' contributions. P.-Y.Q. conceived the research. J.S., Y.Z., L.Q. and C.S. designed this research. J.S., Y.Z. and Y.S. collected the samples. Y.Z., C.C. and J.S. characterized the species list. Y.Z. and R.Z. performed DNA barcoding. Y.H.K., Y.Z., T.W., Y.Y. and J.S. performed population genetics; C.C. performed the EDS analysis. X.W. and L.Y. performed the bathymetric survey. J.S., Y.Z. and C.C. wrote the manuscript. All of the authors read and approved the manuscript.

Competing interests. We have no competing interests.

Funding. This study is part of 'u-loop' ocean ridge hydrothermal vent ecosystem monitoring and conservation project financially supported by China Ocean Mineral Resource R&D Association (COMRA) (DY135-E2-1-03 and DY135-E2-1-02); It is also supported by Hong Kong Branch of South Marine Science and Engineering Guangdong Laboratory (SMSEGL20Sc01) to P.Y.-Q., the Scientific Research Fund of the Second Institute of Oceanography, MNR (grant no. QNYC1902), the National Key Research and Development Program of China (grant no. 2017YFC0306603), a Japan Society for the Promotion of Science Grant-in-Aid for Scientific Research (18K06401) and the Hong Kong Research Grants Council (GRF grant no. 16101219).

Acknowledgements. We thank the captain and crew of the R/V Dayangyihao during the third leg of the COMRA DY52nd cruise, as well as the operation team of the ROV Sea Dragon III. Three anonymous reviewers provided valuable comments that improved an earlier version of this paper.

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
