## [Reviewer comments · Royal Society Open Science]

Review History

RSOS-200110.R0 (Original submission)

Review form: Reviewer 1

Is the manuscript scientifically sound in its present form?

Yes

Are the interpretations and conclusions justified by the results?

Yes

Is the language acceptable?

Yes

Do you have any ethical concerns with this paper?

No

Have you any concerns about statistical analyses in this paper?

No

Recommendation?

Accept with minor revision (please list in comments)

Comments to the Author(s)

The manuscript is well organized and appropriately written to show their findings including 1) updated species list in Tiancheng vent field, 2) population connectivity among hydrothermal vent fields, and 3) variety of mineralization on scales of *C. squamiferum*. These are important to understand deep-sea ecosystem, which is now required as a baseline information for mining related activities in deep ocean. The manuscript is worth publishing as a part of RSOS, but I have a few comments on the manuscript;

The mineralization on the scales of *C. squamiferum* provided us useful information to understand physiology of the snail and biochemical interaction between the snails and hydrothermal fluid. However, this is not well described in the earlier part of the manuscript. It may be helpful for some readers to explain what the authors expected to find by the SEM-EDS analyses, may be in Introduction or in Materials and Methods section.

After the extensive survey in Tiancheng vent field, the authors could not find *R. kairei*, which is one of the representative species in the hydrothermal vent fields in Indian Ocean. The absence of *R. kairei* in Tiancheng vent field, as well as Duanqiao vent field (as reported in the previous study), may be important to characterize fauna in Tiancheng vent field. In addition, as *R. kairei* is distributed in Longqi vent field, the distribution of vent animals is not only controlled by larval dispersal capability, but also by settlement preference, may be caused by environmental difference. The basic information on species distribution may also help to elucidate dispersal and connectivity of faunal communities in vent fields. Although the previous study was not discussed the absence of *R. kairei* in Tiancheng vent field, I think it will improve our understanding in connectivity of vent fauna in Indian Ocean.

Review form: Reviewer 2**Is the manuscript scientifically sound in its present form?**

Yes

Are the interpretations and conclusions justified by the results?

Yes

Is the language acceptable?

Yes

Do you have any ethical concerns with this paper?

No

Have you any concerns about statistical analyses in this paper?

No

Recommendation?

Accept with minor revision (please list in comments)

Comments to the Author(s)

I recommend publication of the manuscript after a few points detailed in my attached report (Appendix A) have been addressed.

Review form: Reviewer 3

Is the manuscript scientifically sound in its present form?

Yes

Are the interpretations and conclusions justified by the results?

Yes

Is the language acceptable?

Yes

Do you have any ethical concerns with this paper?

No

Have you any concerns about statistical analyses in this paper?

No

Recommendation?

Accept with minor revision (please list in comments)

Comments to the Author(s)

A clear and thorough description of faunal assemblages at a recently explored hydrothermal vent field on the SW Indian Ridge, with analysis of faunal similarity with other vent fields, and population genetic structure of three species shared with other vent fields, in the region. Overall, this paper usefully adds to our knowledge of faunal distributions and population connectivity in Indian Ocean hydrothermal vents, which is of interest for fundamental questions in deep-sea species dispersal and biogeography, and also of applied interest for management of potential deep-sea mining in future. I would draw the authors' attention to the specific comments below for minor revisions to improve the clarity of the manuscript even further.

Specific comments on the manuscript:

Throughout: consider changing "Scaly-foot Snail" to "scaly-foot snail", as it is a colloquial name.

Abstract

page 2 lines 42-43: "volatile chimney" - the chimney is not volatile; it is the fluids emitted from it that are volatile. But "volatile" can imply phase-separation of vent fluids into a vapour phase, for which there is no evidence here; "chimney emitting high-temperature vent fluids" would therefore be a more accurate description (and can be inferred from the presence of anhydrite, as reported).

Introduction

page 2 line 57: "Since its discovery... has attracted great attention..." - change to "their discovery" and "have attracted" to match the plural subject of "hydrothermal vents" in the sentence.

page 2 line 1: "Majority of studies..." - add an article (either "A majority..." or "The majority...") for grammatical correctness.

line 32: "...relies on it for nutrition" - should be "relies on them" to match plural bacteria earlier in sentence.

line 38: IUCN status is "Critically Endangered" for scaly-foot snail (rather than just "Endangered").

Methods

page 4 line 36: references [10] and [13] do not contain species presence/absence data in addition to that compiled and analysed in [8] and [16], and so reference to [10] and [13] should be removed here.

Results

Page 6 line 27: "more or less star-burst shaped network" - "more or less" is a vague term and should be removed; "a star-burst shaped network" is an adequate description of the data pattern shown.

Discussion

Page 7 line 36: "Dracogyra subfusca" - the correct species name is "Dracogyra subfuscus"

Figures

Figure 6 caption: When using presence/absence data, rather than abundance data, the equation for the Bray-Curtis similarity index simplifies to that of the Sorensen Index, which predates it; it is therefore Soresen Index that is actually being used here, and the caption should be revised to state that.

Figure 7: no scale bar is presented for the specimen images, which are therefore illustrative rather than for purposes of descriptive morphology etc. They could be presented instead as inserts for each of the haplotype networks in Figure 8 (thereby reducing the number of Figures), as they add little as a separate Figure here.

Decision letter (RSOS-200110.R0)

21-Feb-2020

Dear Dr Sun

On behalf of the Editors, I am pleased to inform you that your Manuscript RSOS-200110 entitled "Nearest vent, dearest friend: Biodiversity of Tiancheng vent field reveals unexpected cross-ridge similarities in the Indian Ocean" has been accepted for publication in Royal Society Open Science subject to minor revision in accordance with the referee suggestions. Please find the referees' comments at the end of this email.

The reviewers and handling editors have recommended publication, but also suggest some minor revisions to your manuscript. Therefore, I invite you to respond to the comments and revise your manuscript.

- Ethics statement

If your study uses humans or animals please include details of the ethical approval received, including the name of the committee that granted approval. For human studies please also detail

whether informed consent was obtained. For field studies on animals please include details of all permissions, licences and/or approvals granted to carry out the fieldwork.

- Data accessibility

If you wish to submit your supporting data or code to Dryad (<http://datadryad.org/>), or modify your current submission to dryad, please use the following link:
<http://datadryad.org/submit?journalID=RSOS&manu=RSOS-200110>

- Competing interests

- Authors' contributions

- Acknowledgements

- Funding statement

Because the schedule for publication is very tight, it is a condition of publication that you submit the revised version of your manuscript before 01-Mar-2020. Please note that the revision deadline will expire at 00.00am on this date. If you do not think you will be able to meet this date please let me know immediately.

If your manuscript is newly submitted and subsequently accepted for publication, you will be asked to pay the article processing charge, unless you request a waiver and this is approved by Royal Society Publishing. You can find out more about the charges at <https://royalsocietypublishing.org/rsos/charges>. Should you have any queries, please contact opscience@royalsociety.org.

Once again, thank you for submitting your manuscript to Royal Society Open Science and I look

forward to receiving your revision. If you have any questions at all, please do not hesitate to get in touch.

on behalf of Dr Punidan Jeyasingh (Associate Editor) and Pete Smith (Subject Editor)
openscience@royalsociety.org

Associate Editor Comments to Author (Dr Punidan Jeyasingh):

Associate Editor: 1

Comments to the Author:

This manuscript was assessed by three experts. All three were uniformly enthusiastic about the manuscript. They make several recommendations to improve the manuscript. I felt the comments were fair and constructive. With much gratitude to the expert reviewers, I invite the authors to make these revisions and resubmit a fresh version.

Reviewer comments to Author:

Reviewer: 1

Comments to the Author(s)

The manuscript is well organized and appropriately written to show their findings including 1) updated species list in Tiancheng vent field, 2) population connectivity among hydrothermal vent fields, and 3) variety of mineralization on scales of *C. squamiferum*. These are important to understand deep-sea ecosystem, which is now required as a baseline information for mining related activities in deep ocean. The manuscript is worth publishing as a part of RSOS, but I have a few comments on the manuscript;

The mineralization on the scales of *C. squamiferum* provided us useful information to understand physiology of the snail and biochemical interaction between the snails and hydrothermal fluid. However, this is not well described in the earlier part of the manuscript. It may be helpful for some readers to explain what the authors expected to find by the SEM-EDS analyses, may be in Introduction or in Materials and Methods section.

After the extensive survey in Tiancheng vent field, the authors could not find *R. kairei*, which is one of the representative species in the hydrothermal vent fields in Indian Ocean. The absence of *R. kairei* in Tiancheng vent field, as well as Duanqiao vent field (as reported in the previous study), may be important to characterize fauna in Tiancheng vent field. In addition, as *R. kairei* is distributed in Longqi vent field, the distribution of vent animals is not only controlled by larval dispersal capability, but also by settlement preference, may be caused by environmental difference. The basic information on species distribution may also help to elucidate dispersal and connectivity of faunal communities in vent fields. Although the previous study was not discussed the absence of *R. kairei* in Tiancheng vent field, I think it will improve our understanding in connectivity of vent fauna in Indian Ocean.

Reviewer: 2

Comments to the Author(s)

I recommend publication of the manuscript after a few points detailed in my attached report have been addressed.

Reviewer: 3

Comments to the Author(s)

A clear and thorough description of faunal assemblages at a recently explored hydrothermal vent field on the SW Indian Ridge, with analysis of faunal similarity with other vent fields, and population genetic structure of three species shared with other vent fields, in the region. Overall, this paper usefully adds to our knowledge of faunal distributions and population connectivity in Indian Ocean hydrothermal vents, which is of interest for fundamental questions in deep-sea species dispersal and biogeography, and also of applied interest for management of potential deep-sea mining in future. I would draw the authors' attention to the specific comments below for minor revisions to improve the clarity of the manuscript even further.

Specific comments on the manuscript:

Throughout: consider changing "Scaly-foot Snail" to "scaly-foot snail", as it is a colloquial name.

Abstract

page 2 lines 42-43: "volatile chimney" - the chimney is not volatile; it is the fluids emitted from it that are volatile. But "volatile" can imply phase-separation of vent fluids into a vapour phase, for which there is no evidence here; "chimney emitting high-temperature vent fluids" would therefore be a more accurate description (and can be inferred from the presence of anhydrite, as reported).

Introduction

page 2 line 57: "Since its discovery... has attracted great attention..." - change to "their discovery" and "have attracted" to match the plural subject of "hydrothermal vents" in the sentence.

page 2 line 1: "Majority of studies..." - add an article (either "A majority..." or "The majority...") for grammatical correctness.

line 32: "...relies on it for nutrition" - should be "relies on them" to match plural bacteria earlier in sentence.

line 38: IUCN status is "Critically Endangered" for scaly-foot snail (rather than just "Endangered").

Methods

page 4 line 36: references [10] and [13] do not contain species presence/absence data in addition to that compiled and analysed in [8] and [16], and so reference to [10] and [13] should be removed here.

Results

Page 6 line 27: "more or less star-burst shaped network" - "more or less" is a vague term and should be removed; "a star-burst shaped network" is an adequate description of the data pattern shown.

Discussion

Page 7 line 36: "Dracogyra subfusca" - the correct species name is "Dracogyra subfuscus"

Figures

Figure 6 caption: When using presence/absence data, rather than abundance data, the equation for the Bray-Curtis similarity index simplifies to that of the Sorensen Index, which predates it; it is therefore Soresen Index that is actually being used here, and the caption should be revised to state that.

Figure 7: no scale bar is presented for the specimen images, which are therefore illustrative rather than for purposes of descriptive morphology etc. They could be presented instead as inserts for each of the haplotype networks in Figure 8 (thereby reducing the number of Figures), as they add little as a separate Figure here.

Author's Response to Decision Letter for (RSOS-200110.R0)

See Appendix B.

Decision letter (RSOS-200110.R1)

27-Feb-2020

Dear Dr Sun,

It is a pleasure to accept your manuscript entitled "Nearest vent, dearest friend: Biodiversity of Tiancheng vent field reveals cross-ridge similarities in the Indian Ocean" in its current form for publication in Royal Society Open Science. The comments of the reviewer(s) who reviewed your manuscript are included at the foot of this letter.

on behalf of Dr Punidan Jeyasingh (Associate Editor) and Pete Smith (Subject Editor)
openscience@royalsociety.org

Appendix A

Report on the manuscript entitled “Nearest vent, dearest friends: Biodiversity of Tiancheng vent field reveals unexpected cross-ridge similarities in the Indian Ocean” by Sun Jin and collaborators.

The manuscript reports the discovery of two new hydrothermal vent sites within a vent field recently described in the eastern part the South West Indian Ridge, as well as an update of the diversity of mega/macrofauna species encountered on this vent field. This updated inventory is used to present a more accurate vision of the distribution of vent fauna across Indian Ridges.

The Indian Ocean remains one of the least explored region on Earth for vent activity and associated faunal communities, and this gap needs to be filled to gain an accurate understanding of vent species connectivity patterns and biogeography. The work presented here is therefore a very useful contribution to the literature on biodiversity at vents in a region where knowledge is particularly lacking, and will contribute to a better understanding of vent biogeography more generally.

The manuscript is clearly presented, easy to follow, and generally presents rather well supported conclusions regarding the closeness of the new vent field to previously known vent fields on the Central Indian Ridge. However, the main message and the title of the paper appear a little overstated, mainly because of weaknesses in the starting assumption.

The initial hypothesis presented by the authors states that the Tiancheng site should harbour faunal communities more similar to those of vents in the western part of the SWIR than to those of the CIR because the RTJ represent a major dispersal barrier between the two ridges. This assumption seems rather strange, and from my point of view, is not the most obvious considering existing knowledge. The authors emphasize the idea that « cross ridge dispersal events are less likely to occur » than dispersal along ridges (favoured by rectified along ridge currents). This is overlooking the fact that between Kairei/Edmond vent fields and Tiancheng, the CIR and SWIR form a quasi-continuum of ridge segments with no major rift offsets, even at the Rodriguez Triple Junction. Much larger distance as well as the presence of several transform faults (Galieni, Atlantis II...) that offset the ridge axis by more than 100 Km between Tiancheng and Longqi/Duanqiao appear much more likely to result in dispersal barriers than the RTJ. A more logical initial hypothesis would be that if barrier to genetic flow exist between CIR vent fields and Longqi/Duanqiao, it is more likely to be in the middle part of SWIR than at the RTJ. The general conclusion of the paper is then not so unexpected.

Evidences presented to support the greater closeness of Tiancheng communities to CIR communities have a few weaknesses. First, nMDS analysis is not convincingly supporting a closer position of Tiancheng to CIR vents, as stated in the discussion. In that regard, I agree more with the sentence in the results that highlights that biological communities at all indian vent fields form a unique cluster, supporting a unique biological province. A strong supporting evidence for closeness between Tiancheng and CIR vent communities, as highlighted by the authors, is the occurrence of species endemic to Longqi/Duanqiao and not found at Tiancheng (and beyond towards the east) like *Gigantopelta*, *Kiwa*, *Dracogyra*, whereas some taxa shared between Tiancheng and CIR vents are not reported at Longqi/Duanqiao, like *Austinograea*, *Alviniconcha*, *Desbruyeresia*. However, regarding the two last species, the data provided is not sufficiently supporting their shared distribution between Tiancheng and CIR vents only. Is the *Desbruyeresia* species reported from Tiancheng, *D. cf marisindica*, actually the same as the species from the CIR? Another *Desbruyeresia* is also reported from Duanqiao (Zhou et al 2018). *Alviniconcha* is known as a cryptic genus with several species (Johnson et al 2015), which reinforce the need to provide genetic evidence that the species found at Tiancheng is actually *A. marisindica*.

My last comment on the general content of the paper is that the analysis of the mineral coating on the scales of *Chrysomallon* gastropods appears a little disconnected from the main part of the

manuscript. In my understanding it is mainly used to highlight how this emblematic species interacts with vent fluids in a different way at Tiancheng than what is reported in other vents, and provides hints regarding geochemical features of Tiancheng vent fluids. Perhaps discussing briefly the role in environmental filtering and local adaptation would provide a link to the general discussion on species distribution in the Indian Ocean.

Generally, the requirement of full data availability is not always met, especially regarding the dataset used for the community structure analysis, and some genetic data.

Overall this paper brings valuable new information on the biodiversity of Indian hydrothermal vents, and I therefore recommend publication in Royal Society Open Science after the main concerns detailed above have been addressed.

Following are a few additional comments and details.

The dataset used for the nMDS community structure analysis is not provided. Although it can be reconstructed from the paper of Zhou et al 2018 that presented the same analysis but without the species newly reported here, it would be easier for the reader to have this dataset provided as a suppl. material.

Accession numbers for COI sequences of *Bathymodiolus marisindicus*, as well as for a number of species listed in table 1 are missing, but I guess they will be provided upon publication of the paper ?

COI sequences were produced for most species, but the number of individuals used is missing (only one ? several ? was potential for cryptic species occurrence assessed ?).

As highlighted by the authors, the *Chrysomallon* gastropod has been listed as an endangered species regarding the deep-sea mining threat: perhaps it should be stated what specific precautions were needed (or not) for collecting this species.

Throughout the manuscript, the authors refer to megafauna when talking about their dataset or biogeography studies published by others. However, this study as well as the cited literature include several macrofaunal species (typically polychaete worms or small limpets). In several instances throughout the ms, megafauna should be replaced with megafauna and macrofauna.

Additional small edits:

Introduction

P1 line 57 : « Since their » instead of « Since its »

P1 lline 58 : « have attracted » instead of « has attracted »

P2 line 35 : « environmental iron ions react » instead of « environmental iron ions reacts »

P2 line 36 : « its habitat is also threatened » instead of « its habitat is also been threatened »

P2 line 49 : « testing » instead of « testifying »

Results

P4 line 38 « *Neolepas marisindica* » instead of « *Neolepas maisindica* »

P4 line60 « with the same fluid source » : I suggest to remove that part of the sentence as the study provides almost no data regarding fluids issuing from the different sites.

Discussion

P6 line 3 « on the surface of its sclerites » instead of « on the surface »

P6 line 60 : « ...planktonic larvae that in one case have » instead of « planktonic larvae that in one case has »

Figure 6 : please provide colour codes, as well as a list of the sites considered for MAR and ESR in the legend.

Figure 9 : A graphic representation of the elemental composition of the mineral deposit layer and of the scale organic matrix would be helpful, perhaps as suppl material.

Figure S1 : It would be nice to provide bathymetric context to help visualise the relative position of the 3 sites.

Appendix B

Associate Editor Comments to Author (Dr Punidan Jeyasingh):

Associate Editor: 1

Comments to the Author:

This manuscript was assessed by three experts. All three were uniformly enthusiastic about the manuscript. They make several recommendations to improve the manuscript. I felt the comments were fair and constructive. With much gratitude to the expert reviewers, I invite the authors to make these revisions and resubmit a fresh version.

Response: We thank the overall positive comment by the associate editor. This manuscript has been thoroughly revised according the comments of these three reviewers.

Reviewer comments to Author:

Reviewer: 1

Comments to the Author(s)

The manuscript is well organized and appropriately written to show their findings including 1) updated species list in Tiancheng vent field, 2) population connectivity among hydrothermal vent fields, and 3) variety of mineralization on scales of *C. squamiferum*. These are important to understand deep-sea ecosystem, which is now required as a baseline information for mining related activities in deep ocean. The manuscript is worth publishing as a part of RSOS, but I have a few comments on the manuscript;

Response: We thank the positive comment by the reviewer.

The mineralization on the scales of *C. squamiferum* provided us useful information to understand physiology of the snail and biochemical interaction between the snails and hydrothermal fluid. However, this is not well described in the earlier part of the manuscript. It may be helpful for some readers to explain what the authors expected to find by the SEM-EDS analyses, may be in Introduction or in Materials and Methods section.

Response: We have included an introductive sentence in the introduction:

‘Different populations have varying levels of metal mineralisation linked to the composition of vent fluid, ranging from none in Solitaire to crystallised iron sulfide in Kairei [21].

And also in the section of “Material and Methods”:

‘Since the minerals that deposit on the sclerites of *Chrysomallon squamiferum* reflects the biochemical reaction of the snail itself with the chemical property of the vent fluid [21], to examine and assess the mineralisation and elemental composition of the sclerites from individuals collected from Tiantang chimney...’

After the extensive survey in Tiancheng vent field, the authors could not find *R. kairei*, which is one of the representative species in the hydrothermal vent fields in Indian Ocean. The absence of *R. kairei* in Tiancheng vent field, as well as Duanqiao vent field (as reported in the previous study), may be important to characterize fauna in Tiancheng vent field. In addition, as *R. kairei* is distributed in Longqi vent field, the distribution of vent animals is not only

controlled by larval dispersal capability, but also by settlement preference, may be caused by environmental difference. The basic information on species distribution may also help to elucidate dispersal and connectivity of faunal communities in vent fields. Although the previous study was not discussed the absence of *R. kairei* in Tiancheng vent field, I think it will improve our understanding in connectivity of vent fauna in Indian Ocean.

Response: We added the discussion about this possibility in the Discussion part, as follows:

‘The absence of the vent shrimp *Rimicaris kairei* in Tiancheng, like Duanqiao [8], is an interesting characteristic since this species is extremely abundant in CIR vent fields and is also known from Longqi but present only in very low abundance [8, 16]. Since Tiancheng is in-between Kairei and Longqi, this indicates that the presence of absence of *R. kairei* is not limited by larval dispersal capabilities but instead the local environment, likely linked to larval settlement preferences.’

Reviewer: 2

Comments to the Author(s)

I recommend publication of the manuscript after a few points detailed in my attached report have been addressed.

Report on the manuscript entitled “Nearest vent, dearest friends: Biodiversity of Tiancheng vent field reveals unexpected cross-ridge similarities in the Indian Ocean” by Sun Jin and collaborators.

Response: We thank the positive comment by the reviewer. The comments in the attached report has been fully considered in the revised version of the manuscript.

The manuscript reports the discovery of two new hydrothermal vent sites within a vent field recently described in the eastern part the South West Indian Ridge, as well as an update of the diversity of mega/macrofauna species encountered on this vent field. This updated inventory is used to present a more accurate vision of the distribution of vent fauna across Indian Ridges.

The Indian Ocean remains one of the least explored region on Earth for vent activity and associated faunal communities, and this gap needs to be filled to gain an accurate understanding of vent species connectivity patterns and biogeography. The work presented here is therefore a very useful contribution to the literature on biodiversity at vents in a region where knowledge is particularly lacking, and will contribute to a better understanding of vent biogeography more generally.

The manuscript is clearly presented, easy to follow, and generally presents rather well supported conclusions regarding the closeness of the new vent field to previously known vent fields on the Central Indian Ridge.

Response: We thank the positive comment by the reviewer.

However, the main message and the title of the paper appear a little overstated, mainly because of weaknesses in the starting assumption. The initial hypothesis presented by the

authors states that the Tiancheng site should harbour faunal communities more similar to those of vents in the western part of the SWIR than to those of the CIR because the RTJ represent a major dispersal barrier between the two ridges. This assumption seems rather strange, and from my point of view, is not the most obvious considering existing knowledge. The authors emphasize the idea that « cross ridge dispersal events are less likely to occur » than dispersal along ridges (favoured by rectified along ridge currents). This is overlooking the fact that between Kairei/Edmond vent fields and Tiancheng, the CIR and SWIR form a quasi-continuum of ridge segments with no major rift offsets, even at the Rodriguez Triple Junction. Much larger distance as well as the presence of several transform faults (Galieni, Atlantis II...) that offset the ridge axis by more than 100 Km between Tiancheng and Longqi/Duanqiao appear much more likely to result in dispersal barriers than the RTJ. A more logical initial hypothesis would be that if barrier to genetic flow exist between CIR vent fields and Longqi/Duanqiao, it is more likely to be in the middle part of SWIR than at the RTJ. The general conclusion of the paper is then not so unexpected.

Response: We agree with the reviewer's comment regarding the transform faults between Longqi/Duanqiao and Tiancheng, and have revised the relevant part of the manuscript. We thank the reviewer for this critical comment which greatly improves the logic of this manuscript.

First, we have removed the word 'unexpected' from the title, as suggested, and added a line about transformation faults in the Abstract.

Second, we have modified the Introduction to include background information on the transformation faults and their potential roles, as you have suggested to us above:

‘An alternative hypothesis involves major transformation faults that are present between Tiancheng and Longqi on the SWIR, such as Galieni and Atlantis II, that offset the ridge axis by more than 100 km between the two vent fields [X]. These may act as significant dispersal barriers and limit the connectivity between Tiancheng and Longqi, while the quasi-continuum between Tiancheng and Kairei may result in the two sites being connected.’

Third, we have modified the Discussion to reflect this. Key modification, among others:

‘Our results therefore indicate that Rodriguez Triple junction is much less significant of a dispersal barrier compared to major transform faults along the SWIR, which may be expected at least for animals considered to disperse through deep currents (e.g., gastropods with lecithotrophic larvae).’

Finally, the Conclusion was also modified as follows:

‘Population genetic results detected substantial population structure between Longqi and Tiancheng but not between Tiancheng and CIR, which challenges the existing hypothesis supporting the hypothesis that the the significant fracture zones between Longqi and Tiancheng on the SWIR present key dispersal barriers for most vent animals but not the between SWIR and CIR is the Rodriguez Triple Junction, enabling and the difficulty for cross-ridge dispersal events to occur between CIR and Tiancheng. These results Instead, our results point to the real dispersal barrier being located in deep water several transformation faults on the SWIR, between Tiancheng and Longqi, warranting future studies to into the specific impacts of each fracture

zone and differing impacts to different taxa with varying dispersal strategies in-
point what it may be.’

We believe these modifications fully address your concerns.

Evidences presented to support the greater closeness of Tiancheng communities to CIR communities have a few weaknesses. First, nMDS analysis is not convincingly supporting a closer position of Tiancheng to CIR vents, as stated in the discussion. In that regard, I agree more with the sentence in the results that highlights that biological communities at all Indian vent fields form a unique cluster, supporting a unique biological province. A strong supporting evidence for closeness between Tiancheng and CIR vent communities, as highlighted by the authors, is the occurrence of species endemic to Longqi/Duanqiao and not found at Tiancheng (and beyond towards the east) like *Gigantopelta*, *Kiwa*, *Dracogyra*, whereas some taxa shared between Tiancheng and CIR vents are not reported at Longqi/Duanqiao, like *Austinograea*, *Alviniconcha*, *Desbruyeresia*.

Response: We can agree, and have modified the nMDS discussion in the Discussion section as follows:

‘The nMDS analysis placed Tiancheng among the other Indian Ocean vents, showing that in a broader scale these vents are all part of a single global vent biogeographic province.’

However, regarding the two last species, the data provided is not sufficiently supporting their shared distribution between Tiancheng and CIR vents only. Is the *Desbruyeresia* species reported from Tiancheng, *D. cf. marisindica*, actually the same as the species from the CIR? Another *Desbruyeresia* is also reported from Duanqiao (Zhou et al 2018).

Response: Regarding *Desbruyeresia cf. marisindica*, because there is no genetic sequence of the samples collected from CIR (type species described), we actually cannot be sure if it is same as the CIR species, you are right. We have therefore modified the sentence to caution the readers:

‘The *Desbruyeresia* species found in Tiancheng is the same as previously reported from Longqi [8] (99% pairwise similarity in COI sequence), although no genetic information is available from the type locality, Kairei.’

Alviniconcha is known as a cryptic genus with several species (Johnson et al 2015), which reinforce the need to provide genetic evidence that the species found at Tiancheng is actually *A. marisindica*.

Response: We only saw *Alviniconcha* snail during our cruise but failed to collect them, and therefore the genetic evidence of this species is lacking. We have added ‘cf.’ when referring to the species. We have added these lines in the Discussion to highlight this:

‘Although geographically the *Alviniconcha* in Tiancheng is most likely *A. marisindica*, we were unable to collect specimens and must refer to it as *Alviniconcha cf. marisindica* for the time being.’

‘Collection of specimens and genetic information are required to identify

and examine the connectivity of *Alviniconcha* population in Tiancheng with those on the CIR, which would be an important focus for future visits to the vent.’

My last comment on the general content of the paper is that the analysis of the mineral coating on the scales of *Chrysomallon* gastropods appears a little disconnected from the main part of the manuscript. In my understanding it is mainly used to highlight how this emblematic species interacts with vent fluids in a different way at Tiancheng than what is reported in other vents, and provides hints regarding geochemical features of Tiancheng vent fluids. Perhaps discussing briefly the role environmental filtering and local adaptation would provide a link to the general discussion on species distribution in the Indian Ocean.

Response: We added a sentence to the Introduction to fill in background on the scaly-foot snail:

“Different populations have varying levels of metal mineralisation linked to the composition of vent fluid, ranging from none in Solitaire to crystallised iron sulfide in Kairei [21].

Furthermore, we included a sentence on the potential role of environmental filtering and local adaptation in the end of Discussion to tie together the scaly-foot snail mineralisation results with the rest (this links in nicely after a sentence added for *Rimicaris* shrimps added following suggestions by Reviewer 1):

‘The different mineralisation present on the scales of the Tiancheng scaly-foot snail compared to other sites may reflect such differences in the local environment, especially in the vent fluid. In ‘extreme’ environments such as hydrothermal vents, taxa are often locally adapted to a narrow range of environments and the environment may in turn select for which taxa may be present in environmental filtering, which may be an important factor determining species distributions in the Indian Ocean hot vents. Understanding the interaction between biology and the environment in this sense is of great importance in biological conservation of these sites, for example protection of sites with a wide range of environment variation may be required in order to conserve the full species diversity.’

Generally, the requirement of full data availability is not always met, especially regarding the dataset used for the community structure analysis, and some genetic data.

Response: The dataset used for the community structure analysis is now included as Supplementary Table.

We have now included all of the genetic accession numbers used in this manuscript, in order to comply with data availability requirements.

Overall this paper brings valuable new information on the biodiversity of Indian hydrothermal vents, and I therefore recommend publication in Royal Society Open Science after the main concerns detailed above have been addressed.

Response: We thank the positive comments by the reviewer.

Following are a few additional comments and details.

The dataset used for the nMDS community structure analysis is not provided. Although it can be reconstructed from the paper of Zhou et al 2018 that presented the same analysis but without the species newly reported here, it would be easier for the reader to have this dataset provided as a suppl. material.

Response: the raw data of nMDS is now included in the Supplementary materials.

Accession numbers for COI sequences of *Bathymodiolus marisindicus*, as well as for a number of species listed in table 1 are missing, but I guess they will be provided upon publication of the paper ?

Response: We have now included all missing accession number for genetic sequences.

COI sequences were produced for most species, but the number of individuals used is missing (only one ? several ? was potential for cryptic species occurrence assessed ?).

As highlighted by the authors, the Chrysomallon gastropod has been listed as an endangered species regarding the deep-sea mining threat: perhaps it should be stated what specific precautions were needed (or not) for collecting this species.

Response: The collecting for this paper has actually happened before the IUCN Red List inclusion, and therefore no precautions were needed. Added a line to specify this: ', after the collecting for the present study has taken place'

Throughout the manuscript, the authors refer to megafauna when talking about their dataset or biogeography studies published by others. However, this study as well as the cited literature include several macrofaunal species (typically polychaete worms or small limpets). In several instances throughout the ms, megafauna should be replaced with megafauna and macrofauna.

Response: This has been revised throughout the manuscript.

Additional small edits:

Introduction

P1 line 57 : « Since their » instead of « Since its »

Response: This has been revised.

P1 lline 58 : « have attracted » instead of « has attracted »

Response: This has been revised.

P2 line 35 : « environmental iron ions react » instead of « environmental iron ions reacts »

Response: This has been revised.

P2 line 36 : « its habitat is also threatened » instead of « its habitat is also been threatened »

Response: This has been revised.

P2 line 49 : « testing » instead of « testifying »

Response: This has been revised.

Results

P4 line 38 « *Neolepas marisindica* » instead of « *Neolepas maisindica* »

Response: this has been revised.

P4 line60 « with the same fluid source » : I suggest to remove that part of the sentence as the study provides almost no data regarding fluids issuing from the different sites.

Response: this part of the sentence has been removed.

Discussion

P6 line 3 « on the surface of its sclerites » instead of « on the surface »

Response: this has been revised.

P6 line 60 : « ...planktonic larvae that in one case have » instead of « planktonic larvae that in one case has »

Response: this has been revised.

Figure 6 : please provide colour codes, as well as a list of the sites considered for MAR and ESR in the legend.

Response: the colour code was included.

Figure 9 : A graphic representation of the elemental composition of the mineral deposit layer and of the scale organic matrix would be helpful, perhaps as suppl material.

Response: We have now included graphs showing EDS Net Counts of both layers in the Supplementary Material.

Figure S1 : It would be nice to provide bathymetric context to help visualise the relative position of the 3 sites.

Response: We unfortunately do not have detailed bathymetric around the three sites. We only have a rough bathymetric at bigger scale which was illustrated in Figure 1b showing the overall position of the Tiancheng vent field.

Reviewer: 3

Comments to the Author(s)

A clear and thorough description of faunal assemblages at a recently explored hydrothermal vent field on the SW Indian Ridge, with analysis of faunal similarity with other vent fields, and population genetic structure of three species shared with other vent fields, in the region. Overall, this paper usefully adds to our knowledge of faunal distributions and population connectivity in Indian Ocean hydrothermal vents, which is of interest for fundamental questions in deep-sea species dispersal and biogeography, and also of applied interest for

management of potential deep-sea mining in future. I would draw the authors' attention to the specific comments below for minor revisions to improve the clarity of the manuscript even further.

Response: We thank the reviewer's positive comment, and the following comments are properly addressed or clarified.

Specific comments on the manuscript:

Throughout: consider changing "Scaly-foot Snail" to "scaly-foot snail", as it is a colloquial name.

Response: Changed throughout the manuscript as suggested.

Abstract

page 2 lines 42-43: "volatile chimney" - the chimney is not volatile; it is the fluids emitted from it that are volatile. But "volatile" can imply phase-separation of vent fluids into a vapour phase, for which there is no evidence here; "chimney emitting high-temperature vent fluids" would therefore be a more accurate description (and can be inferred from the presence of anhydrite, as reported).

Response: This has been revised as suggested.

Introduction

page 2 line 57: "Since its discovery... has attracted great attention..." - change to "their discovery" and "have attracted" to match the plural subject of "hydrothermal vents" in the sentence.

Response: This has been revised.

page 2 line 1: "Majority of studies..." - add an article (either "A majority..." or "The majority...") for grammatical correctness.

Response: This has been revised to the "The majority of".

line 32: "...relies on it for nutrition" - should be "relies on them" to match plural bacteria earlier in sentence.

Response: This has been revised.

line 38: IUCN status is "Critically Endangered" for scaly-foot snail (rather than just "Endangered").

Response: Scaly-foot snail is indeed "Endangered" species and not "Critically Endangered", please check the official IUCN red list website:

<https://www.iucnredlist.org/species/103636217/103636261>

Methods

page 4 line 36: references [10] and [13] do not contain species presence/absence data in addition to that compiled and analysed in [8] and [16], and so reference to [10] and [13] should be removed here.

Response: There is no reference [13] in page 4 line 36. We used parts of formerly published data for the nMDS analysis, which includes data from SWIR (Ref. 8), Eastern Scotia Ridge (Ref. 10), Longqi (Ref. 16), and also CIR (Ref. 26). That is why 10 is also listed here.

Results

Page 6 line 27: "more or less star-burst shaped network" - "more or less" is a vague term and should be removed; "a star-burst shaped network" is an adequate description of the data pattern shown.

Response: This has been removed.

Discussion

Page 7 line 36: "Dracogyra subfusca" - the correct species name is "Dracogyra subfuscus"

Response: the correct name is indeed *Dracogyra subfusca*, with specific epithet ending modified since the original description due to incorrect gender agreement as per ICZN.

Figures

Figure 6 caption: When using presence/absence data, rather than abundance data, the equation for the Bray-Curtis similarity index simplifies to that of the Sorensen Index, which predates it; it is therefore Soresen Index that is actually being used here, and the caption should be revised to state that.

Response: This has been revised.

Figure 7: no scale bar is presented for the specimen images, which are therefore illustrative rather than for purposes of descriptive morphology etc. They could be presented instead as inserts for each of the haplotype networks in Figure 8 (thereby reducing the number of Figures), as they add little as a separate Figure here.

Response: We would like to keep this image as we believe it is important to show the morphology of esp. scaly-foot snail, and we are under no limitation for figure numbers in the present manuscript. We have included approximate size measurements of the specimens in the figure captions, we hope this is ok.